

# Combining traditional and novel techniques to increase our understanding of the lock-in depth of atmospheric gases in polar ice cores - results from the EastGRIP region

Julien Westhoff[1], Johannes Freitag[2], Anaïs Orsi[3,4], Patricia Martinerie[5], Ilka Weikusat[2,6], Michael Dyonisius[1], Xavier Faïn[5], Kevin Fourteau[5], and Thomas Blunier[1]

[1]Niels Bohr Institute, University of Copenhagen, Copenhagen, Denmark
[2]Alfred Wegener Institute, Helmholtz Centre for Polar and Marine Research, Bremerhaven, Germany
[3]Laboratoire des Sciences du Climat et de l'Environnement LSCE/IPSL, CEA-CNRS-UVSQ, Universite Paris-Saclay, Gif-sur-Yvette, France
[4]The University of British Columbia, Department of Earth, Ocean and Atmospheric Sciences, Vancouver, Canada
[5]Univ. Grenoble Alpes, CNRS, INRAE, IRD, Grenoble INP, IGE, Grenoble, France
[6]Department of Geosciences, Eberhard Karls University Tübingen, Germany

**Correspondence:** Julien Westhoff (julien.westhoff@nbi.ku.dk)

**Abstract.** We investigate the lock-in zone (LIZ) of the EastGRIP region, Northeast Greenland, in detail. We present results from the firn air pumping campaign of the S6 borehole, forward modeling, and a novel technique for finding the lock-in depth (LID, the top of the LIZ) based on the visual stratigraphy of the EastGRIP ice core. The findings in this work help to deepen our knowledge of how atmospheric gases are trapped in ice cores. $CO_2$, $\delta^{15}N$, and $CH_4$ data suggest the LID lies around 58 to 61 m depth. With the grayscale and bright spot analysis based on visual stratigraphy, we can pinpoint a change in ice properties to exactly 58.3 m depth, which we define as the optical lock-in depth (OLID). This visual change in ice properties is caused by the formation of rounded and enclosed air bubbles, altering the measured refraction of the light pathways. The results for the LID and OLID agree accurately on the depth. We furthermore use the visual stratigraphy images to obtain information on the sharpness of the open to closed porosity transition. Combing traditional methods with the independent optical method presented here, we can now better constrain the bubble closure processes in polar firn.





## 1 Introduction

### 1.1 Trapping Gases in Ice Cores

Ice cores, from polar ice sheets, provide a rare opportunity to directly measure the composition of the air from our past atmosphere far back in time (Petit et al. 1999). To determine the age of the trapped air precisely, it is necessary to know the age difference between the air bubbles and the surrounding ice, the so-called delta age. As snow accumulates on top of the polar ice sheets, it densifies into firn and ice (Herron et al. 1980). In the upper firn column, the air in the interstitial space between the snow grains is still connected to the open atmosphere. As the snow grains compact under the weight of overlying accumulation, the air in the interstitial space becomes isolated from the open atmosphere, slowly trapped, and occluded into the bubbles.

With regards to gas, the firn column is divided into three zones (Sowers et al. 1992). Blunier et al. (2000) describe three zones derived from the succession of $\delta^{15}N$ of the atmosphere: an upper convection zone, a diffusive zone, and a lock-in zone (LIZ). A small number of bubbles are already closed-off in the diffusive zone. However, the bulk of bubbles is closed-off in the LIZ. The top of the LIZ, the lock-in-depth (LID) is defined as the depth at which the $\delta^{15}N$ becomes constant (in the case of stationary firn) and marks where diffusivity essentially drops to zero. The close-off depth (COD) at the bottom of the LIZ is where essentially all bubbles are closed.

### 1.2 Sites Locations

The East Greenland Ice Core Project (EastGRIP) and S6 sites lie approximately one kilometer apart, inside the Northeast Greenland Ice Stream (NEGIS) in Northeastern Greenland (EastGRIP coordinates: 75°38'N, 36°00'W, 2704 m a.s.l., Vallelonga et al. 2014). The EastGRIP site has a fairly stable accumulation rate of 0.11 m ice eq. yr$^{-1}$ over the past 400 years (Vallelonga et al. 2014) and an annual mean temperature of $-28.7°C$ (Vandecrux et al. 2023). The surface velocity at the sites is approximately 55 m/a (Hvidberg et al. 2020), and the coordinates provided here refer to the 2018 location. The firn air sampling campaign was conducted on the S6 borehole (75°37'14.0"N, 35°58'16.1"W).

Solely for investigating density, we include data from the S2- and NEGIS-ice cores which are located in the EastGRIP region, as well as the North Greenland Eemian Ice Drilling (NEEM) ice core from northwestern Greenland (77.45°N 51.06°W).

### 1.3 Motivation

In this study, we investigate the firn-ice transition using traditional methods and a novel technique based on the physical properties of the ice core, to precisely determine the depth at which individual bubbles are closed-off from the surrounding firn. For this work, we define this depth of physical closure of individual bubbles as the optical lock-in depth (OLID). We will show that the OLID coincides with the LID determined with traditional methods. Therefore, we present the results from the firn-air pumping campaign and modeling results to reconstruct the LID and COD from the S6 firn core.



This study is divided into two parts. Part I presents and discusses estimations of the LID at EastGRIP estimated from traditional methods. Part II focuses on the new line scan-based methodology to find the OLID at EastGRIP and discusses the benefits of this technique to study the firn-ice transition.

## 2    Part I: Lock-in Depth determination from gas profiles in firn air

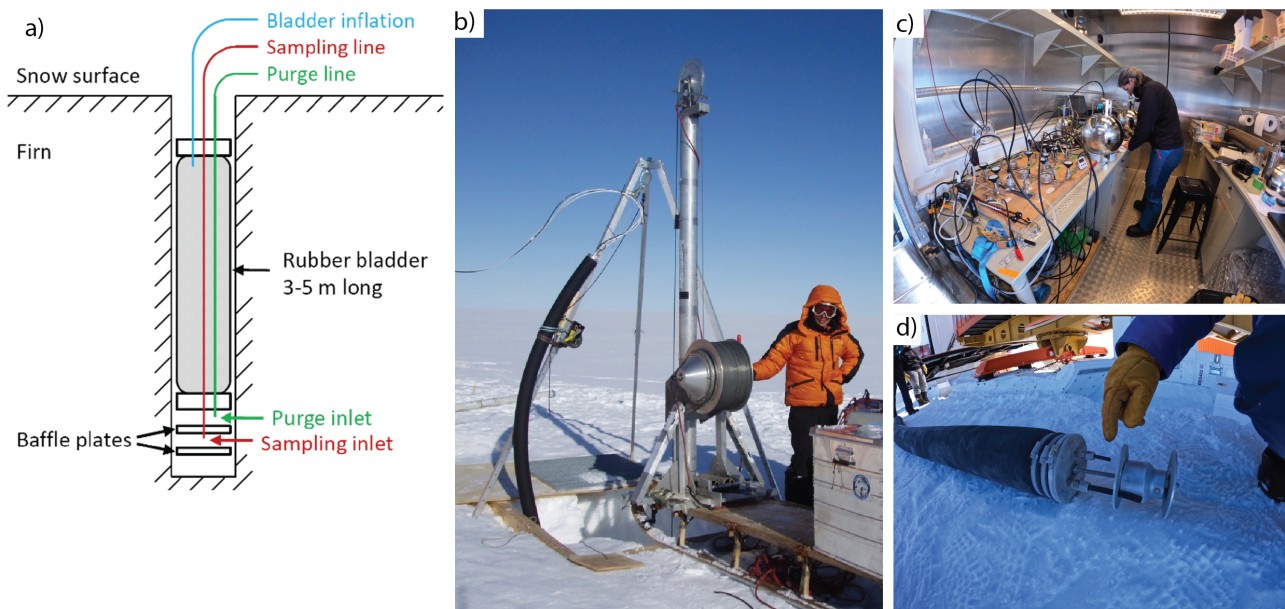

**Figure 1.** a) Sketch of the firn air pumping setup, b) firn air pumping in the field, c) sampling of the pumped air at Little Dome C, Antarctica 2022, and d) the purge and sampling inlets on the actual setup.

### 2.1    Variation Between Cores

When analyzing and comparing the results from two different locations, we must keep in mind that spatial variations can occur. At the NEEM site, Buizert et al. (2012) describe two shallow cores which are only 64 m apart from each other. Despite the close proximity of these two shallow cores, they show differences in measured mixing ratio profiles that exceed the experimental error and the COD differs vertically by approximately half a meter.

   Riverman et al. (2019) and Oraschewski et al. (2021) suggest that ice inside NEGIS is subjected to enhanced densification

from the sides, due to flow, and not only vertically by gravity.

   Nevertheless, we compare results from different ice cores to use the maximum amount of data available for our analysis.





## 2.2 Firn air pumping at the S6 site

In order to determine how the composition of interstitial air evolves with depth until it is trapped, we conducted a firn air campaign June 12-21, 2018, in the S6 borehole.

We successively drilled a segment of ice core, typically three to ten meters, and then removed the drilling equipment, and inserted a "bladder" into the hole. The three-meter-long natural rubber bladder (fig. 1b) seals the bottom of the hole from the modern air, letting solely a 1.4" purge line and a 3/8" sampling line pass through (fig. 1a and d). These lines are connected to the surface (fig. 1c) and a pump is used to inflate the bladder and suck air from the bottom of the borehole.

The $CO_2$ and $CH_4$ compositions of the firn air were monitored on-site to detect issues and provide a first determination of the age of the air. Four to seven flasks were filled at each depth, to later measure the air composition in more detail. The total pumping time at each depth was 1.5 to 3.5 hrs depending on the gas volume extracted.

In the field, the pumping conditions are stable throughout the diffusive zone, with similar, slightly decreasing concentrations of $CO_2$ and $CH_4$, mirroring the atmospheric increase of the past decades. The rapid decrease in $CO_2$ and $CH_4$ concentrations below 60.86 m are the first indications that the LIZ is reached, as the air suddenly ages rapidly, and diffusive mixing essentially stops. Soon after, the sampling system's flow rate decreases, until filling the sampling canisters becomes impractical. That depth is then called the sampling close-off depth. At EastGRIP, it was reached at 66.61 m.

## 2.3 Results from the S6 borehole

Trace gas measurements in flasks filled during the firn air pumping at EastGRIP provide direct information about the location of the LIZ. We present the information obtained from the structure of $\delta^{15}N$, $CO_2$, and $CH_4$ mixing ratios in the open porosity of the firn. Moreover, $CH_4$ measurements in the closed porosity were also retrieved from CFA analysis of the deep firn and young ice, providing additional constraints on gas trapping in bubbles. Methane data in the open and closed porosity of the firn were used to constrain the evolution pore closure at EastGRIP using a model of gas transport in firn (Witrant et al. 2012).

### 2.3.1 $\delta^{15}N$ isotopic ratios

The $\delta^{15}N$ ratio is constant in the atmosphere over millions of years (Mariotti 1983). It is therefore a great indicator of the fractionation processes happening in the firn. $\delta^{15}N$ is reported using modern atmospheric air as a reference. First, mixing prevents isotopic fractionation in the convective zone, leaving $\delta^{15}N$ at zero (same as the air). Then, in the diffusive zone, nitrogen isotopes fractionate: 1) due to gravitational settling, heavy isotopes are concentrated in the bottom of the firn, showing a linear increase of $\delta^{15}N$ with depth. 2) Due to thermal gradients, heavy isotopes concentrate in the cold portion of the firn. In fig. 2a, we can see the impact of the seasonal temperature minimum from the previous winter as an extremum in $\delta^{15}N$, near 12m. Finally, in the bottom section of the firn, the enrichment of $\delta^{15}N$ stops, while there is still enough open pore space to extract air. This absence of gravitational enrichment is evidence that vertical diffusion has stopped in this zone, although there is still a significant amount of open porosity. It is likely due to the layered nature of the firn: a dense layer is sealed off above a more porous layer, from where air can be extracted, but this air is no longer in diffusive equilibrium with the open firn





**Figure 2.** a) $\delta^{15}N$ (not corrected for thermal fractionation) and b) $CO_2$ measurements from the firn air pumping campaign on the S6 borehole, indicating the transition of the diffusive to non-diffusive zone to lie just above, or just below, 60 m depth, respectively. c) $CH_4$ measured in the ice core, as a function of depth, around the bubble close-off. d) Forward modeling of $CH_4$ for closed porosity. e) Laboratory measurements of $CH_4$ in the air from the open porosity of firn (symbols) compared with IGE-GIPSA firn model results obtained with the two open/closed porosity ratio parameterizations shown in d (the two model results are nearly superimposed). f) Open/closed porosity ratio parameterizations used in the IGE-GIPSA firn model, blue: usual parameterization using Equation (6) in Witrant et al. 2012), green: the exponential factor in Eq. (6) from Witrant et al. 2012) is reduced from 7.6 to 3.0.



column. This last zone is the LIZ. The slight decrease of $\delta^{15}N$ in the LIZ can be due to either a small contribution of thermal fractionation or a small contamination with modern air. Indeed, as we pump rather heavily the layer above when the bladder is deflated, the deep firn fills up with modern air that can then contaminate the deeper sample, although care is taken in the field to mitigate this effect. The maximum of $\delta^{15}N$ at 60.86 m gives an approximate value for the LID (fig. 2a).

### 2.3.2 $CO_2$ mixing ratios

Over the course of the past two centuries, the atmospheric $CO_2$ mole fraction, commonly referred to as concentration, has almost doubled to approximately 420 ppm today (https://gml.noaa.gov/ccgg/trends/). The depth range of the firn air pumping reaches a depth of 66 m, corresponding to almost 400 years of snow deposition. In the diffusive zone, the firn air is in contact with the atmosphere, and stays young: the $CO_2$ concentration remains close to the atmospheric values, and decreases slowly. In the lock-in zone, vertical diffusion is impeded, and the air ages as fast as the ice, which causes the $CO_2$ concentration to decrease rapidly with depth.

The slope of the $CO_2$ measurements from the S6 borehole-firn air-pumping campaign, significantly changes at a depth of 58 m (fig. 2b). This change of slope suggests that the diffusivity is now rapidly decreasing and the top of the LID has been reached.

### 2.3.3 $CH_4$ mixing ratios in open and closed pores

$CH_4$ mixing ratios in the open porosity of the firn (dots in fig. 2e) were measured at IGE (Institut des Géosciences de l'Environnement) and show an overall similar behavior as $CO_2$, with a slope change in the 58-60 m depth range indicating the start of the LIZ.

In addition, the S6 firn core was measured by the continuous flow analysis (CFA) method for methane (Chappellaz et al. 2013, Fain et al. 2022). Partial close-off results in intrusion of modern air either since the time of core recovery or, in case the pores remain open until the time of measurement, during the measurements. Fourteau et al. (2019) observed correlated variations of density, methane mixing ratio, and liquid conductivity in the LIZ of an Antarctic ice core, with positive spikes of methane induced by laboratory air reaching the center of the sample occurring in minimum density layers (fig. 9 in Fourteau et al. 2019). Figure 2c shows a typical behavior of sharply decreasing $CH_4$ variability with depth in the LIZ, reflecting the diminishing occurrence of sharp methane peaks due to modern air intrusions (fig. 2c).

Layers with not fully closed pores can be found down to 71.5 m while from around 66 m the bubbles are essentially closed off. Open porous layers alternate between impermeable layers essentially hindering gas transport below the lock in depth. We note that such gas transport occurring at the centimeter scale in the CFA sample may not have an impact on the overall firn scale as the porosity in the surrounding layers is mostly closed. Methane data from CFA measurements in deep firn are too variable above 65 m to be used as a predictor of the LID but provide very good indications about full air entrapment in the closed porosity which occurs above 72 m depth with some local layering effects.




### 2.3.4 Model constraints on open to total porosity ratio

Progressive bubble closure during firn densification results in a gradual decrease in the ratio of open to total porosity (values from one to zero) and the complete trapping of air into bubbles. Firn models simulating gas entrapment in ice usually represent the open/total porosity ratio with a simple parameterization as a function of density (e.g. Goujon et al. 2003; Buizert et al. 2012). This ratio is difficult to measure and available data show inconsistencies (Fourteau et al. 2019; Fourteau et al. 2020 and references therein). Here we adjust the open/total porosity ratio parameterization in a model of gas transport in firn (Witrant et al. 2012) in order to best simulate the $CH_4$ concentrations in the closed porosity of the EastGRIP S6 core. Note that EastGRIP is a complex site, and constant temperature and accumulation for the stationary firn model might be offset for EastGRIP due to the effect of strain-densification (Riverman et al. 2019; Oraschewski et al. 2021). The usual parameterization (Witrant et al. 2012, eqs. 4 and 6, blue in fig. 2f), adapted from Goujon et al. (2003) and also used in Buizert et al. (2012) results in too high $CH_4$ concentrations in the model versus the data in the closed porosity of firn (fig. 2d). In order to improve the model to data match we modified the open/total porosity ratio parameterization of Witrant et al. (2012), eqs. 4 and 6 by changing the exponential factor from 7.6 to 3.0 (green curve in fig. 2f). A better match of the $CH_4$ CFA data is obtained (fig. 2d, in green), while the $CH_4$ data in the open porosity are correctly modeled in both simulations (fig. 2e) because a firn diffusivity tuning was performed for each simulation (Witrant et al. 2012). The beginning of the bubble closure moves significantly upwards resulting in 50% closed porosity at around 61.3 m (green in fig. 2f) compared to around 64.8 m in the basic (blue in fig. 2f) configuration.

### 2.4 Summary of Part 1

We use $\delta^{15}N$, $CO_2$, and $CH_4$ (fig. 2a,b,e) data from well-established methods to estimate the top depth of the LIZ. The depths we find are between 58 and 61 m, for the S6 borehole site. These depth ranges spread over 3 m and with an annual layer thickness of 11 cm, this already results in an uncertainty of approximately 27 years. We also find two end-member scenarios of how the porosity of the firn/ice matrix changes from open to closed (fig. 2d). Both end-members equally satisfy the firn air $CH_4$ data, although the green curve is better tuned to the ice core $CH_4$ concentrations. Together, these results show that firn air concentrations do not provide an unequivocal horizon for the LID, or the evolution of closed porosity.

These results leave room for a more exact determination of where gases are locked-into bubbles and where diffusivity approaches zero. The closing of pores is a physical property of the firn/ice matrix, and measurements of physical properties should help constrain the bubble closure process. In the following section we investigate how optical scanning of the ice core can provide information on the bubble closure process, addressing these questions: Is it possible to find the LID or COD using images obtained from the optical dark field line scanner? Can we find a single solution for the gradual change from open to closed porosity? How sharp is this transition, which later determines the range of gas ages in the ice?



## 3 Part II: Inspecting Physical Properties to learn more about the firn-ice transition

By obtaining information about the physical properties of firn and ice from density measurements and line scan data, we aim to deepen our knowledge of how gases are trapped in the ice matrix. We introduce a new method to precisely confine the LIZ and thus shine more light on the firn-ice transition. We use this method to also obtain insight into the transition from open to closed porosity.

### 3.1 Density measurements from four Greenland ice cores

Density measurements have been conducted on the S2, NEGIS, and NEEM ice cores, which are all located in central Greenland. The density increase follows a similar trend in all three cores (fig. 3a). The NEEM ice core starts higher and density increases slower than in the other two cores. This trend is due to different boundary conditions at its location 400 km away from the other two cores on the other side of the Greenland ice divide. The S2 and NEGIS cores are located close together (see site location

section) and both exceed the value of $790 kg/m^3$ at around 60 m depth.

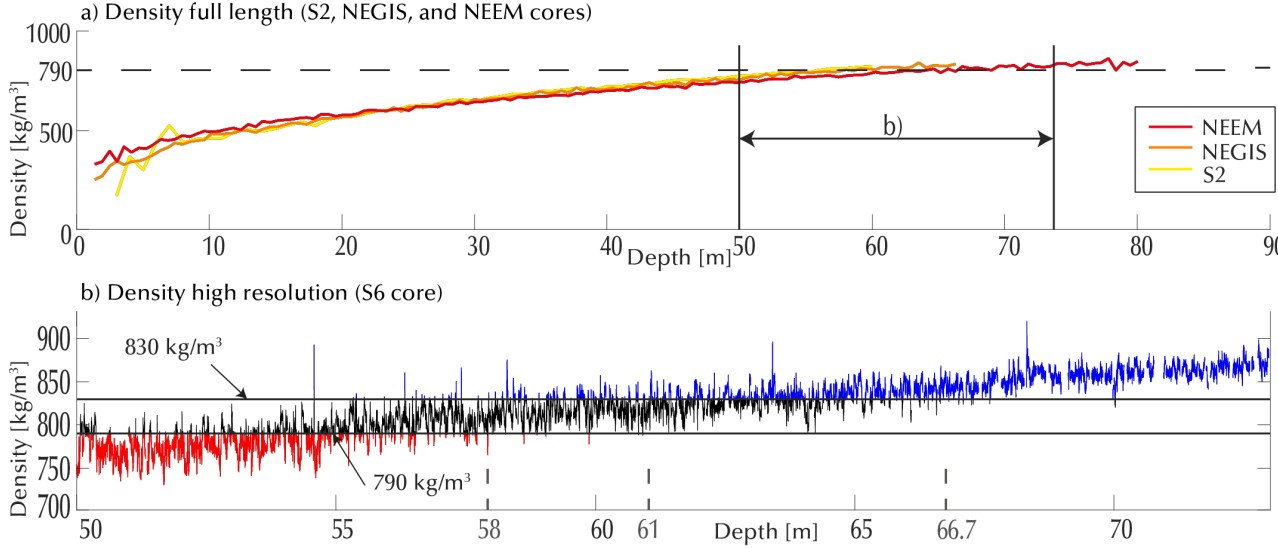

**Figure 3.** a) Density of the S2, NEGIS, and NEEM ice cores, as 1 m, 55 cm, and 55 cm averages, respectively, measured by weighing in the field. b) High-resolution density of the S6 ice core derived from micro-CT measurements. Values below $790 kg/m^3$ in red, values above $830 kg/m^3$ in blue, and values in-between in black. Depth results from part 1 in gray on the x-axis.

In general, the close-off zone is defined by a density between $790 kg/m^3$ and $830 kg/m^3$ (Schwander et al. 1984). At a density of $830 kg/m^3$ the material is defined as ice and bubbles are believed to be completely closed-off.

High-resolution density data from micro-CT scanning is available in a depth range from 50 to 73 m, conducted on the S6 core (fig. 3b). With depth, the density increases and exceeds $830 kg/m^3$ the first time at 54.58 m depth. This depth corresponds

to a melt layer found in the EastGRIP core (Westhoff et al. 2022). Below this, we find three peaks at 56.33, 57.42, and 58.3 m





depth, which are high-density layers and could potentially be a seal for gas diffusion. Between 55 and approximately 67 m depth, density values lie between $790kg/m^3$ and $830kg/m^3$, suggesting this to be the LIZ. Below this depth, essentially all density values correspond to bubbly ice.

## 3.2  Visual Stratigraphy

### 165  3.2.1  Line scanning ice cores

The line scanner images a polished, 165 cm-long, ice core slab from above, while the firn/ice is being illuminated from below (Svensson et al. 2005; Faria et al. 2018; Westhoff et al. 2020). This causes light to be reflected on various features, such as snow and firn grains, core breaks, bubbles, dust-rich layers, and others. In ice from the Holocene period, corresponding to the upper approximately 1100 m of the EastGRIP ice core, the main cause of reflection, refraction, and scattering are air bubbles

within the ice and core breaks (Westhoff et al. 2022). These features then appear as bright sections in the line scan.

On most ice cores the line scanning is only done below the firn, i.e. in solid ice. The only two where the firn section has also been scanned are EastGRIP and Renland Ice Cap (ReCAP) ice cores. We thus have a unique chance to investigate the firn section using the line scan images. For this study, we focus on the EastGRIP line scan images.

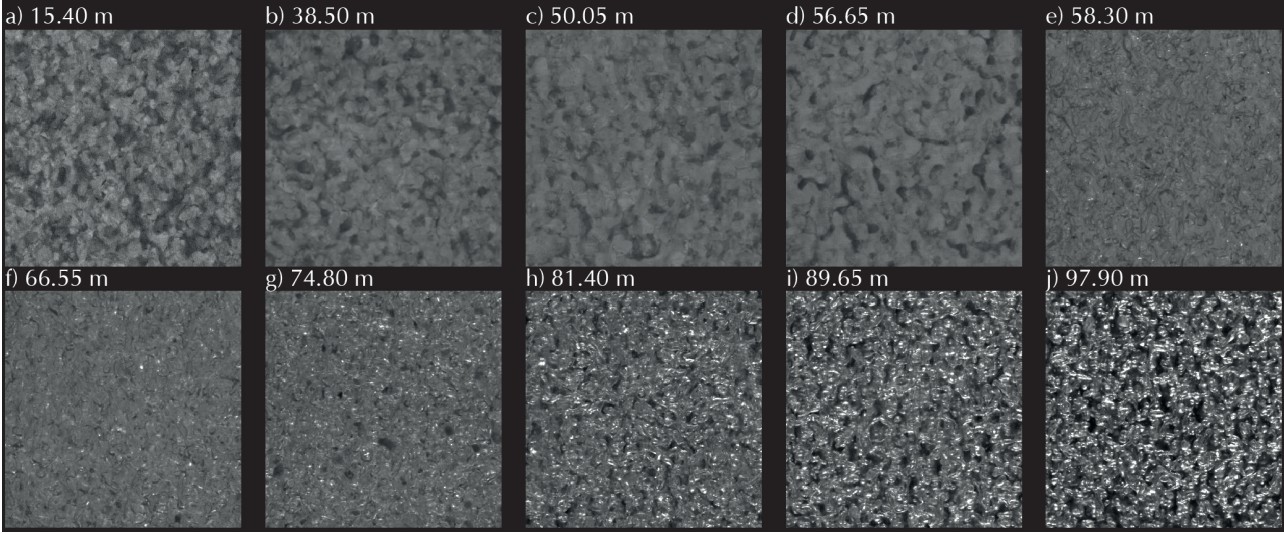

**Figure 4.** Two by two-centimeter images from the line scanner. Examples of the optical appearance in line scanner image, with enhanced brightness for visualization. Spots appearing bright by the eye have pixel values above 60.

### 3.2.2  The Visual Stratigraphy Images

Between firn and ice, there is an inversion of bright and dark sections (Westhoff et al. 2022). Low-density firn and snow can be seen as ice grains embedded in a matrix of air: the grains thus appear bright as they are small objects scattering light (fig. 4a





to d). In dense bubbly ice, the picture is reversed: we have bubbles embedded in an ice matrix, which is now the bright objects scattering light (fig. 4h to j). In the transition from firn to ice, we see mixed elements of both appearances (fig. 4e to g). In the supplement of this work, we provide a detailed description, explanation, and interpretation of the images in fig. 4.

### 3.3 Density and Visual Stratigraphy derived lock-in depth

The minimum pixel values (fig. 5a, yellow line) of each $1 \times 5\ cm^2$ increment remain relatively constant over the upper 100 meters of the EastGRIP ice core. This ensures constant camera settings without any adjustments made to the brightness. From the first processed ice core section (13.75 m) to 58 m, minimum (yellow), median (red), and maximum values (brown) do not vary much and maximum values are low (fig. 5a). We see a small peak at 50 m, and two troughs at around 54 and 58 m. Below 58 m, we see a trend of linearly increasing maximum pixel values (fig. 5a, dashed brown line) from 58 m to 85 m. During this increase, the median value (red) remains at a constant value, indicating that there are very few bright pixels in one increment. Below 85 m depth, the maximum values remain at 255 over several samples. We identify a drop at around 90 m and then the value remains at a maximum of 255. During the increase of pixel values below 58 m depth, we identify peaks, indicating some sort of layering in this section (discussed later). Figure 5a has a resolution of one $1 \times 5\ cm^2$ sample every 55 cm (in the center of every bag).

Figure 5b is an attempt to determine the number of bubbles from the bright spot proxy. Between 58 and 85 m, we see an exponential increase (purple dashed line) in bright spots. Between 85 and 90 m, we find two drops to very low bright spot numbers and then a sudden increase to between 800 and 1200 bright spots per $1 \times 5\ cm^2$ image, where it remains constant with depth.

In the interval between 53.35 to 69.85 m, we inspected the bubble closure in detail, with a continuous analysis, i.e. one increment every centimeter (fig. 5d, pink box in fig. 5a). The drops of all pixel values around 54 m and 58 m (trough 54 and trough 58, respectively) have a width of approximately 30 cm. Below the last trough, at 58.30 m, the maximum pixel values (brown) greatly increase, almost reaching the maximum of 255. Higher maximum pixel values than the average of shallow depths represent the first appearance of bright spots in the line scan images (see also fig. 4e). The detailed inspection does not show a gradual increase in maximum pixel values, but rather multiple peaks and troughs, representing layers containing more and fewer bubbles.

Figure 5e (green box in fig. 5b) also shows the sudden appearance of bright spots at a depth of 58.3 m. The peaks in fig. 5d and e align, but peaks are more visible in d. The number of bright spots does not show a gradual increase, but just like fig. 5d, multiple peaks and troughs. A prominent example of this alternation is peak B (fig. 5e) where we find a prominent trough followed by a peak.

From the density data, we calculate the total porosity of our ice core samples (Witrant et al. 2012, eq. 2 and 3). The S2, NEGIS, and NEEM total porosity gradually decreases over the measured depth (fig. 5c). The high-resolution total porosity data, derived from micro-CT density data from the S6 core, shows four cases of porosity greatly dropping between 54 and 59 m depth (fig. 5f). A very dense layer at exactly 58.3 m depth, corresponding to very low porosity, coincides with the depth at which we find the first bright spots. This could be the impermeable layer, below which diffusion is increasingly limited.





**Figure 5.** Depth of investigation from 13.75 m to 99.40 m below the surface with one measurement every 55 cm (a and b). Detailed analysis from 53.35 to 69.85 m, with continuous measurements every centimeter (d and e). Our analysis locates the OLID at 58.30 m depth. We find a linear increase in maximum values in a) and a roughly exponential increase in bubbles in b) (dashed lines). Both d) and e) show a sudden appearance of bright spots below 58.30 m with layering well visible in e). c) Total porosity, calculated from density data of the S2, NEGIS, and NEEM ice cores, and f) total porosity from micro-CT density data of the S6 ice core (full high-resolution S6 profile will be published in Freitag et al. (in prep.).





## 3.4 What causes these bright spots in our images?

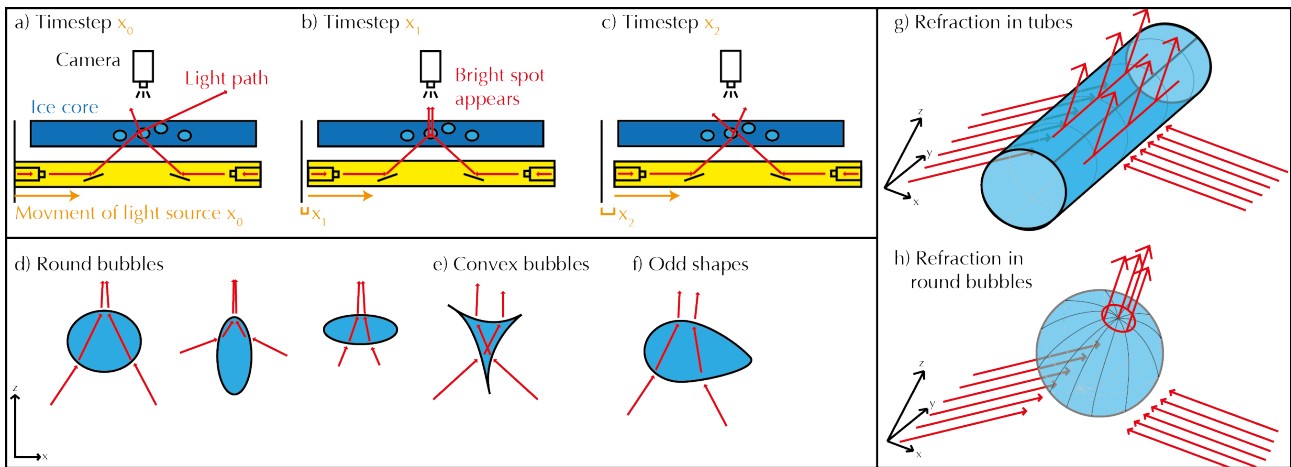

**Figure 6.** a) to c) Line scan setup with a fixed camera and ice core, and a moving light source (yellow). In certain settings the introduced light will refract and bundle, causing bright spots (b). The amount of light bundling is a geometric effect, dependent on the bubble shape, e.g.: d) rounded bubbles, e) convex bubbles, or f) other shapes. The amount of bundling is also dependent on the 3D bubble shape: tubes bundle less (g) than spheres (h).

The termination of the pore space structure into isolated clusters and eventually into individual spherical bubbles is a percolation transition. If we think of the pore space as a pore network with many redundant connections, then, above a certain density, the connections gradually disappear, and isolated clusters of different sizes are formed while a permeable network also remains. With depth, i.e. in the process of metamorphism, many connections are cut and the permeable clusters become increasingly smaller. The LID is the horizon where pore clusters become disconnected with the surface. Below that, the isolated pore clusters disintegrate down to the smallest units: bubbles (i.e. closed-off bubbles), upon further compaction. This percolation transition is superimposed on the stratification, i.e. layers with different percolation states alternate. Therefore, large percolation clusters can still be found below the LID and air can be extracted from them.

If we assume that bright spots are created by singular/spherical bubbles, we do not expect them to occur at the beginning of the percolation transition, but only when most connections are closed off. It therefore makes sense that we do not see the bright spots at the first occurring closed porosity (around 50 m), but further down.

Our analysis shows bright spots below a depth of 58.3 m. We interpret these bright spots to be a geometric effect of rounded bubbles causing refraction of the injected light, bundling it, and causing this bright appearance in the image (fig. 6). With the correct bubble shape, these bubbles will then appear bright as the light source moves below the ice core (fig. 6a to c) We assume that these bright spots only occur once bubbles are rounded or have an ellipsoid shape, are sealed off, and trapped in the ice matrix (fig. 6d,h). Clusters would not create bright spots with such intensity (fig. 6h). Odd shapes of bubbles will most likely not lead to such an intense bundling of light and therefore appear less bright (fig. 6f).





## 3.5 Stratigraphic layering in the lock-in zone

The idea that layering in the LIZ can influence the closure of bubbles has been around for a long time (e.g. Blunier et al. 2000; Fourteau et al. 2019; Mitchell et al. 2015). This idea motivated Birner et al. (2018) to establish a 2D model which includes layering, in order to better reproduce the lock-in of air bubbles. With the line scan images of the EastGRIP firn section, we now have a unique chance to further investigate this layering.

The pore network is not homogeneous, as temperature gradient metamorphism in the upper snowpack already introduces anisotropy into the structure. The strangulation does, therefore, not progress continuously with the reduction of porosity but rather collapses in phases. One of these phases seems to be reached when the impermeable cluster collapses and single pores separate.

Figure 5a shows that between 58 and 85 m depth, more and more layers contain these bright spots until, below 85 m depth, all image increments have at least one bright spot, suggesting all layers to be impermeable. The data used for fig. 5a,b are obtained from one $1 \times 5\ cm^2$ increment every 55 cm. Therefore, the plot does not show an average of many values, but one single snapshot of that depth. We see strong variations in the maximum pixel values as we do not always measure a layer with rounded bubbles and bright spots.

Figure 5d,e show the variability during the lock-in. Some layers have bright spots, while others appear darker (d), going hand in hand with the number of bubbles we find (e). Thus, some layers don't have these rounded, ellipsoidal, and sealed-off bubbles while others do. An exact correlation of bubble number (fig. 5e) with density (fig. 5f) remains a challenge, as the analysis was conducted on different ice cores, EastGRIP and S6, respectively.

From the visual stratigraphy data, the (optical) lock-in seems to appear suddenly. Yet we must keep in mind that a small change from odd shape clusters to rounded bubbles can cause a significant change in optical appearance (fig. 5d to f, green bar). The number of bright spots (fig. 5e) shows a small increase at first and layers with more and others with fewer bubbles are visible. This indicates layering, as not all odd-shaped clusters transform into rounded bubbles at the same depth. When the bubble number drops to zero, the reason is mostly missing data due to core breaks. Yet the height of the peaks, i.e. the number of bright spots, varies throughout the LIZ.

## 3.6 Ice core porosity derived from visual stratigraphy

### 3.6.1 Results

From the fraction of open to total porosity (fig. 2f), we calculate the fraction of closed to total porosity. $closed/total = 1 - open/total$. The two end-member states are displayed in green and blue (fig. 7). Results from Part 1 suggest 100% closed porosity at a depth of 67 m (dashed vertical line in fig. 7).

Using line scan image data, we calculate the percentage of pixel values exceeding 20 of the total image (an explanation of why exactly 20 is given in the supplement). Above 40 m depth, almost all pixels have values below 20, i.e. the ratio of bright pixels to total pixels is 0% (black curve in fig. 7). At a depth of around 46 m the ratio of brighter to total pixels increases



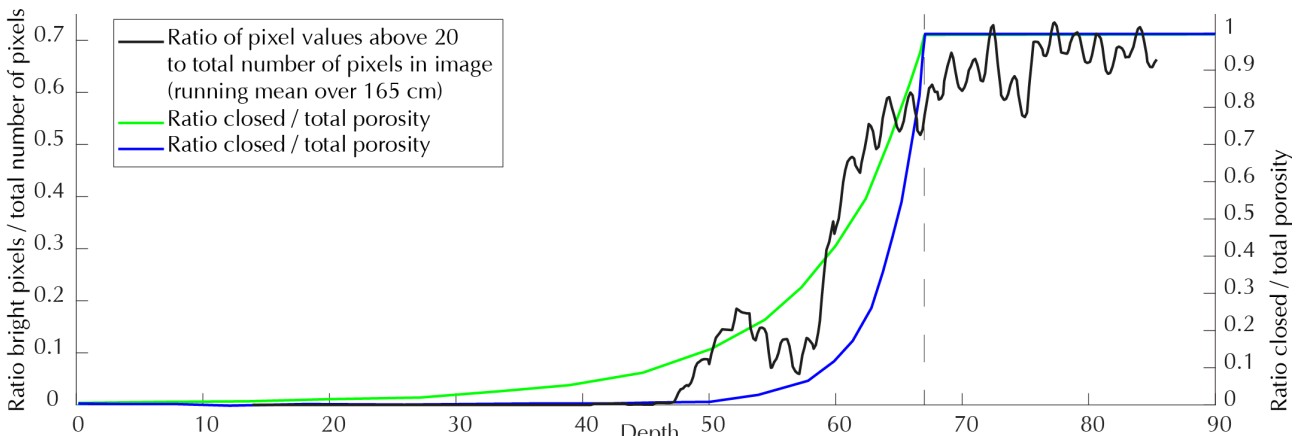

**Figure 7.** Closed/total porosity calculated from open/total porosity (fig. 2b) in green and blue. Percent of the image covered by pixels greater than 20 in black. Running means over one line scan sample (165 cm) to reduce brightness fluctuations within one sample.

slightly and sharply increases at a depth of 58.3 m. The ratio gradually plateaus at a depth of 70 m at values around 65 to 70%. Due to the nature of line scan images, a ratio of approximately 0.7 is not exceeded and can be considered the maximum.

### 3.6.2 Discussion

We note here that using a pixel cut-off of 20 brightness units is a simple way to describe a more complicated process. As
the metamorphism of firn grains to ice crystals takes place, their structure significantly changes. Firn, in the line scan images, appears as a homogeneous surface with dark patches in between, the open pores (supplement fig. D1). Closed pores appear differently, i.e. brighter, sharper brightness contrasts, and with apparently more structure (fig. D1b, just above the star). This closure adds a third feature to the visual stratigraphy image, now containing ice, open, and closed pores. This increase in structural complexity can be seen in fig. D1c, as thin lines in the line scan image. The appearance of ice, without open pores,
is visible in the absence of any homogeneously bright image patches, i.e. former firn. While the process of transforming open pores to closed pores is a complex process, it can easily be described by analyzing the number of pixels in an image exceeding a value of 20.

    The ratio of pixel values above 20 to the total number of pixels (black line in fig. 7) shows the trend of how the firn/ice-air interfaces are changing due to pores becoming closed-off and thus becoming more reflective for the light of the line scanner.
The pixel ratio (black curve) has a strong similarity to the ratio of closed to total porosity (green curve) and both follow the same trend. The ratio of closed to total porosity increases with depth, until all pores are closed-off at a depth of approximately 67 m. This process is more gradual, over many meters around 67 m, in the line scan data (black) and shows similarity to the work of Mitchell et al. (2015) who measure the closed porosity directly.





Although the line scan data is shown as a running mean average over 1.65 m, we still see layering and changes above the annual variability. This indicates further layering in the firn, e.g. with more closed-off bubbles from 50 to 55 m and fewer from 55 to 60 m depth (fig. 7).

### 3.7 Summary of Part 2

By using physical properties data, obtained by density measurements and visual stratigraphy data, we can reveal more details about the firn-ice transition. High-resolution density data show the presence of high-density, or low-porosity, layers exactly at the depth where airflow starts to reduce to a minimum during the firn air pumping campaign. Visual stratigraphy pinpoints the exact depth, where a change in physical properties causes the appearance of bright spots. This sudden appearance of bright spots is associated with a change in the firn/ice to air boundaries and the shape of bubbles. The number of bright spots as a function of depth indicates a layering in the firn and ice, favoring some layers to contain more bright spots, i.e. closed-off bubbles, compared to others.

By analyzing the change of appearance of firn and pores with depth, we find a simple correlation between closed pore space and image brightness. Using this correlation, we can add more details to the transition of open to closed porosity throughout the firn-ice transition.

Including physical properties data to the traditional methods of defining the LIZ, can add details and can further increase our understanding of the firn-ice transition.

## 4 Conclusion

All results presented in this work, point to the transition of diffusive to non-diffusive zone in the EastGRIP-area to lie at approximately 58 to 61 m depth. $\delta^{15}N$ suggests a depth of 61 m (fig. 2a), $CO_2$ and $CH_4$ a depth of around 58 m (fig. 2b and e, respectively). The OLID suggests a depth of 58.3 m (fig. 5d). Thus, all methods, including the optical method, agree very well.

The ice from 58.3 m depth was deposited in the year 1681 CE (Mojtabavi et al. 2020), i.e. 338 years before the 2018 drilling. Including a typical mixing delay of $\pm$ 10 to 50 years of Greenland sites (Schwander et al. 1988) the gas-ice age difference ranges from 288 to 328 years.

In this work, we have presented a method based on the physical properties of the ice core to obtain information about the LIZ, the transition from diffusive to non-diffusive zone, and to increase our understanding of open to closed porosity. The method works accurately and is in agreement with the traditional methods which obtain results from firn air pumping campaigns in boreholes. It further offers the advantage of resolving centimeter-scale variability, which is beneficial to better understand the impact of firn stratification on gas entrapment. We have investigated the EastGRIP area, as we had the best data coverage from this region. We hope to use this method on sites around Greenland and Antarctica in the future.





## 5 Outlook

The importance for future line scan processing is to have a continuous camera setting, unchanged throughout the upper 100 m of the ice core. Hereby two scans should be performed: a dark setting, to make use of our method, and a bright setting to make the images analyzable by eye.

### 5.1 ReCAP, coastal Greenland

The ReCAP ice core, from eastern coastal Greenland, is the only other core where visual stratigraphy data are available from
315 the ice sheet surface to below the firn/ice transition. When trying to analyze this core, we faced many challenges, such as changes in camera settings, numerous melt layers disturbing the light reflections, and strong changes in brightness between summer and winter layers. A first test shows that it is possible to use this method on the ReCAP ice core, despite the challenges. We plan to analyze the RECAP core in the future, together with other ice cores which will have been line scanned by then.

### 5.2 BeyondEPICA, Antarctica

The BeyondEPICA (European Project for Ice Coring in Antarctica) ice core is currently being drilled. Processing has not taken place yet and we hope to acquire line scan images throughout the firn section. With these images, we can then test the here-presented method on an Antarctic ice core, where conditions are expected to be more similar to EastGRIP than to ReCAP, as we do not expect melt layers.



**Appendix A: Firn-air Data Collection**

Firn air sampling took place approximately one kilometer out of the EastGRIP camp between June 12 to 21, 2018. We drilled with the Copenhagen 3-inch shallow drill to a depth just below 66 m. The ice core retrieved is called the 'S6 firn core'. After partial drilling, the French firn air sampling system was lowered into the hole, and the bladder, to seal the hole air-tight, was inflated. Air was then extracted between the bladder and the current bottom of the borehole (Schwander et al. 1993). These flask samples were taken at 22 depth levels from the surface to 66 m depth. In parallel to the flask samples, a Picarro cavity ring-down instrument (Type G1301) was used to monitor the concentrations of $CO_2$ and $CH_4$. Before the instrument the air was pulled through a magnesium perchlorate trap to remove water vapor, making the water vapor correction obsolete. Although the measurements have been calibrated with our working standard, their purpose was monitoring during the sampling process. The data do not have the same quality as laboratory measurements, thus, some offsets are to be expected. The flow rate during sampling was about 4.3 l per minute down to 61 m below the surface. The flow rate then decreased to below 1 l per minute at 66 m depth where we stopped sampling. The low flow rate at the lowest level indicates that the pores in the firn are almost closed-off.

**Appendix B: The Technique in Detail - Grayscale and Bubble Analysis**

The grayscale analysis of line scan images is commonly done on recent ice cores (e.g. Svensson et al. 2005; Morcillo et al. 2020, Stoll et al. 2023). We perform a grayscale analysis on $1x5\ cm^2$-increments by measuring the intensity values for pixels (px) in the selected area. We measure the minimum, median, and maximum values of each increment. The OLID is based on the changes of these values with depth. One centimeter in our line scan sample is represented by 186 px, each increment thus has a size of 186 by 930 px. The pixel values range from 0 to 255, representing 256 possibilities. The value of the single lowest/highest pixel inside this 186 by 930 px-area is then used to describe the minimum/maximum value of this depth.

We investigate the uppermost section of the EastGRIP ice core, i.e. from 13.75 m below the surface to 99.40 m (examples shown in fig. 4). We measure one $1x5\ cm^2$ sample every 55 cm, i.e. every bag, for the upper 100 meters. For 16 meters around the LID, we measure continuous, i.e. without gaps, $1x5\ cm^2$ samples for higher resolution. From each line scan image, we remove two centimeters from the top, around sections with core breaks, and the bottom. The removed data points are sampling artifacts, obscuring the brightness.

The lighting from below the ice core slab is refracted and bundled on these rounded solid-gas interfaces causing bright reflections, white spots, or, using the terminology from Morcillo et al. (2020), bright spots. We count the bright spots as a proxy for the number of bubbles in our samples. Bright spots are clusters of bright pixels in a binary image. To convert the grayscale image (pixel range from 0 to 255) to a binary image (0 or 255), we use three different thresholds: 60, 150, and 250. We do not use the method of Ueltzhöffer et al. (2010) due to complicated 3D effects from our thick line scans. Neither do we use a blur function (or similar) to correct for gaps in bright pixels belonging to the same bright spot and might therefore overestimate the number of bubbles. Yet, our analysis is simple and sufficient to detect the first formation of bubbles and also makes layering visible within the LIZ.





## Appendix C:  Detailed Analysis of EastGRIP Line Scan Images

Line scan images provide a 2D picture of the structure of the ice cores. The focus depth is set to a few millimeters below the ice core's polished surface. While these 2D images cannot provide a full 3D reconstruction such as e.g. a computer tomography scan (e.g. Freitag et al. 2013 or Lipenkov 2018), it does reveal some structures and we can get a hint of what the 3D structure could look like. Experience in using the line scanner reveals that when using the camera settings as done in fig. 4, we have approximately half a centimeter in focus, which is then visible in the image. As light is introduced at an angle from below, the ice below and above our focus depth remains invisible.

Figure 4a (depth below the ice sheet surface: 15.40 m) shows one of the first samples that was scanned. We see firn grains in different shadings of gray and voids scattered across the two-by-two-centimeter image. From fig. 4a to d, the firn grains remain bright and the voids connect to more elongated structures.

Our image analysis shows a significant change in the appearance of texture at 58.30 m depth (between fig. 4d, e). This change is represented by the first appearance of white spots in the images, which are evidence of the first formation of rounded continuous ice-air interfaces, i.e. trapped air bubbles. The lighting from below the ice core slab is reflected on these rounded solid-gas interfaces causing bright spots.

Further analysis of sections below the firn-ice transition shows an increase in bright spots, which reveals the gradual closure of air pathways. As this is a 2D analysis, connections into the y-axis (into the page) can easily be missed. Yet, analyzing the x- and z-axis (lateral extension and depth-axis, respectively) gives an idea of the distribution and scale of air pathways.

While fig. 4e shows bright spots, these are optically not visible as closed bubbles but could also be pathways evolving into a rounded shape. The first definite evidence of closed bubbles can be found at a depth of 66 m, where on a 2D appearance some bubbles appear fully enclosed (fig. 4f).

At a depth of 74.8 m (fig. 4g) we can identify closed bubbles, together with tubes of bubbles, which are still connected. These tubes and bubbles decrease in size, causing a further closure of pathways within the ice (fig. 4h).

At a depth of 89.65 m (fig. 4i) all bubbles are fully closed off, without any sub-cm-scale pathways on the x- and z-axis. Pathways may still appear in the y-direction. While bubbles form, the space of the image occupied by dark areas, i.e. clean ice, starts to increase.

## Appendix D:  Why cut off at 20 pix for closed/total porosity plot

When the firn porosity is still fully open, e.g. in a depth of 44 m, then the firn and the open pores have pixel values around 15 and 10, respectively (fig. D1a). With depth, the black patches in the images remain at a pixel value of 10 (fig. D1). As bubbles close-off, their edges become brighter and gradually exceed a pixel-value of 20 (fig. D1b to c). At increasing depth the air-ice interfaces appear brighter, exceeding a pixel value of 20, and single bright spots become visible (fig. D1d). The pixel value of 20, therefore, seems to be a reasonable cut-off, of when firn transitions to ice and the reflectivity increases. Note, that these values are dependent on the camera settings and need to be adjusted for other ice cores and/or sets of images.



**Figure D1.** The change of brightness appearance with depth. a) above the firn-ice transition, b-c) the transition, and d) below the firn-ice transition. The left side shows the brightness profiles from the blue lines on the right side. Image width is approximately 2 cm.



## Appendix E: Not the effect of microtoming ice core surface

**Figure E1.** Brightness enhanced images over the optical LID (EastGRIP). a) and b) show the appearance of firn without bright spots (here converted to black), c) the transition, and d) multiple bright spots and clearer firn-gas interfaces.

In an early stage of this work, the sudden appearance of bright spots in the line scan data was allocated to a change in ice core processing, i.e. starting to microtome the ice core surface before scanning it. We investigate $1x5\ cm^2$ images over the optical LID to find that the appearance of bright spots is an effect of metamorphism, and not of a change in ice core processing procedure. The transition to locked-in bubbles, and our proposed OLID, appear in fig. E1b and c at 58.3 m depth. Between fig. E1a/b and d, we see a distinct difference in appearance, as firn-to-gas boundaries become more distinct. Figure E1c shows both appearances in one image, on the left we see an appearance similar to b), and on the right similar to d). As we see both



appearances in one image, the effect of bright spot occurrence can not be due to the start of microtoming the surface but is an effect of metamorphism.

The black pixels in fig. E1c and d represent over-saturated pixels, i.e. those detected by our bright spot analysis.

## Appendix F: Melt Layers Around the Lock-in Depth

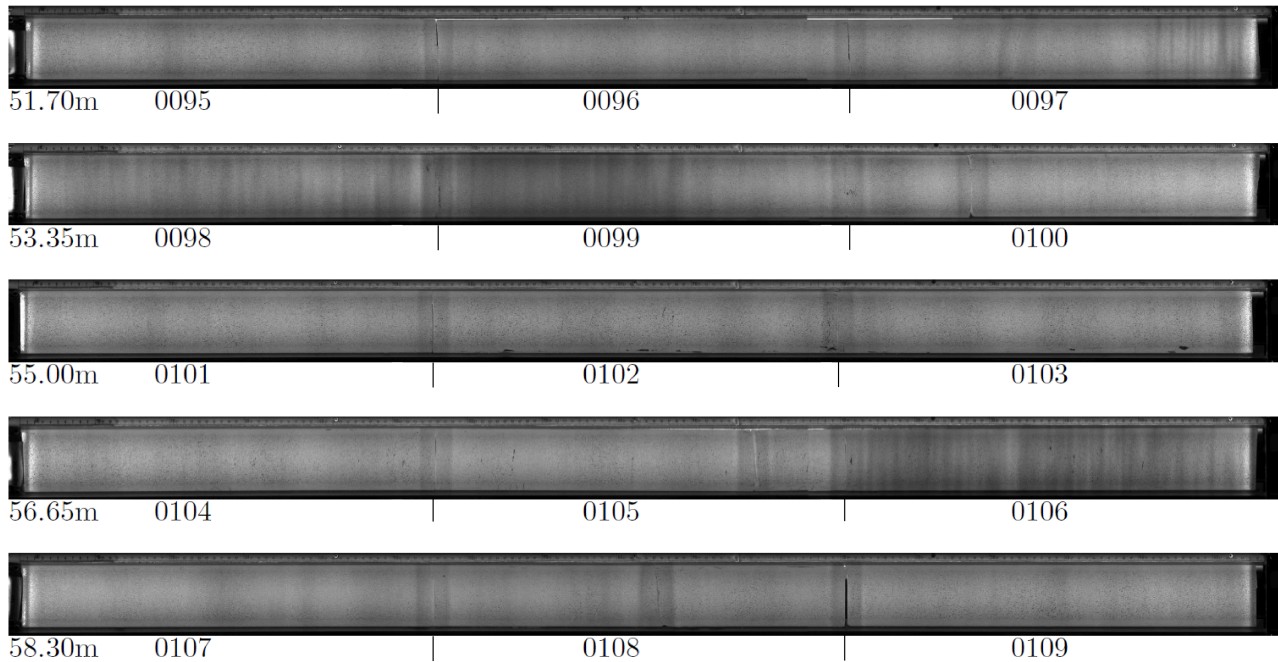

**Figure F1.** Bag 95 to 109, showing brightness variations in the line scan images and melt layers (from overview booklet Westhoff et al. 2019).

The study by Westhoff et al. (2022) provides an overview of melt events in the EastGRIP ice core. In the following, we analyze the melt layers and -lenses in the proximity of the LID. As Blunier et al. (2000) state: "All mixing processes are strongly influenced by the presence or absence of icy layers resulting from surface melting during summer". In the 16 m around the OLID (fig. 5d,e) we find two melt layers in total: at 54.80 and 68.60 m depth. The upper one should be treated with caution, as it is located next to a core break and could be an imaging artifact, rather than a true layer (top of bag 99, fig. F1,
Westhoff et al. 2022). In the following 30 cm below this melt layer (bag 99), we find the section with significantly darker line scan images (i.e., lower grayscale values, compare to fig. 5e, trough 54). Darker layers such as these have been influenced by summer surface melting, or pre-melting (Dash et al. 2006).

The ice in 55 to 58 m depth is characterized by the occurrence of eleven melt lenses in total. These are small melt patches with a horizontal extent less than the core width, i.e. less than 10 cm. Four of these lie between 58.00 and 58.16 cm depth



(fig. 5e), just above the LID and inside the darker line scan image section (bag 106, fig. F1). Melt lenses are evidence of strong surface melting, most likely during summer, and thus an alteration of the physical properties of the snowpack. This alteration can work as a block for vertical diffusion and thus enhance the formation of the first bubbles below this depth.

The influence of a melt layer at 68.6 m is visible in our data (fig. 5e), but not to an outstanding degree. Just below the melt layer, we find a layer containing many bubbles, probably an effect of reduced diffusivity due to the solid, bubble-free ice layer

above. Yet this peak (peak C, fig. 5e) is equally high as other bubble-rich layers and thus indicates that melt layers must not necessarily cause an outstanding effect on bubble close-off. A reason for this uncertainty could be the limited size, 10 cm width, of the ice core and image, and therefore the inability to quantify the extent of the melt layer over more than ten centimeters.



*Author contributions.* JW came up with the initial idea of using the optical method, wrote most of the manuscript, and led the work. JF contributed with density data, observations, ideas on how the optical method works, and sections in part 2. AO, PM, XF, KF, and TB

contributed with $\delta^{15}N$, $CO_2$, and $CH_4$ data, its interpretation, firn-air pumping insights (AO), the modeling results (PM and XF), and wrote sections in part 1. IW contributed with the line scan data and interpretations on the firn-ice transition. MND and TB contributed with general context and combining the optical and traditional data.

*Competing interests.* The authors declare that no competing interests are present

*Acknowledgements.* EastGRIP is directed and organized by the Centre for Ice and Climate at the Niels Bohr Institute, University of Copen-

425 hagen. It is supported by funding agencies and institutions in Denmark (A. P. Møller Foundation, University of Copenhagen), USA (US National Science Foundation, Office of Polar Programs), Germany (Alfred Wegener Institute, Helmholtz Centre for Polar and Marine Research), Japan (National Institute of Polar Research and Arctic Challenge for Sustainability), Norway (University of Bergen and Trond Mohn Foundation), Switzerland (Swiss National Science Foundation), France (French Polar Institute Paul-Emile Victor, Institute for Geosciences and Environmental research), Canada (University of Manitoba) and China (Chinese Academy of Sciences and Beijing Normal University).

Julien Westhoff thanks the Villum Foundation, as this work was supported by the Villum Investigator Project IceFlow (NR. 16572). Support for this work was also provided in France by the CNRS INSU LEFE program.



# Appendix: References

Birner, B., C. Buizert, T. J. Wagner, and J. P. Severinghaus (2018). "The influence of layering and barometric pumping on firn air transport in a 2-D model". In: *Cryosphere* 12.6, pp. 2021–2037. ISSN: 19940424. DOI: 10.5194/tc-12-2021-2018.

Blunier, T. and J. Schwander (2000). "Gas enclosure in ice: age difference and fractionation". In: *Physics of Ice Core Records*, pp. 307–326. URL: http://eprints.lib.hokudai.ac.jp/dspace/handle/2115/32473.

Buizert, C. et al. (2012). "Gas transport in firn: Multiple-tracer characterisation and model intercomparison for NEEM, Northern Greenland". In: *Atmospheric Chemistry and Physics* 12.9, pp. 4259–4277. ISSN: 16807316. DOI: 10.5194/acp-12-4259-2012.

Chappellaz, J., C. Stowasser, T. Blunier, D. Baslev-Clausen, E. J. Brook, R. Dallmayr, X. Fain, J. E. Lee, L. E. Mitchell, O. Pascual, D. Romanini, J. Rosen, and S. Schüpbach (2013). "High-resolution glacial and deglacial record of atmospheric methane by continuous-flow and laser spectrometer analysis along the NEEM ice core". In: *Climate of the Past* 9.6, pp. 2579–2593. DOI: 10.5194/cp-9-2579-2013. URL: https://cp.copernicus.org/articles/9/2579/2013/.

Dash, J. G., A. W. Rempel, and J. S. Wettlaufer (2006). "The physics of premelted ice and its geophysical consequences". In: *Reviews of Modern Physics* 78.3, pp. 695–741. ISSN: 15390756. DOI: 10.1103/RevModPhys.78.695.

Fain, X., R. H. Rhodes, P. Place, V. V. Petrenko, K. Fourteau, N. Chellman, E. Crosier, J. R. McConnell, E. J. Brook, T. Blunier, M. Legrand, and J. Chappellaz (2022). "Northern Hemisphere atmospheric history of carbon monoxide since preindustrial times reconstructed from multiple Greenland ice cores". In: *Climate of the Past* 18.3, pp. 631–647. DOI: 10.5194/cp-18-631-2022. URL: https://cp.copernicus.org/articles/18/631/2022/.

Faria, S. H., S. Kipfstuhl, and A. Lambrecht (2018). *The EPICA-DML Deep Ice Core*. Berlin: Springer-Verlag GmbH Germany. ISBN: 9783662553060.

Fourteau, K., P. Martinerie, X. Fain, C. F. Schaller, R. J. Tuckwell, H. Löwe, L. Arnaud, O. Magand, E. R. Thomas, J. Freitag, R. Mulvaney, M. Schneebeli, and V. Y. Lipenkov (2019). "Multi-tracer study of gas trapping in an East Antarctic ice core". In: *The Cryosphere* 13.12, pp. 3383–3403. DOI: 10.5194/tc-13-3383-2019. URL: https://tc.copernicus.org/articles/13/3383/2019/.

Fourteau, K., F. Gillet-Chaulet, P. Martinerie, and X. Fain (2020). "A micro-mechanical model for the transformation of dry polar firn into ice using the level-set method". In: *Frontiers in Earth Science* 8, p. 101.

Freitag, J., S. Kipfstuhl, and T. Laepple (2013). "Core-scale radioscopic imaging: A new method reveals density-calcium link in Antarctic firn". In: *Journal of Glaciology* 59.218, pp. 1009–1014. ISSN: 00221430. DOI: 10.3189/2013JoG13J028.

Goujon, C., J.-M. Barnola, and C. Ritz (2003). "Modeling the densification of polar firn including heat diffusion: Application to close-off characteristics and gas isotopic fractionation for Antarctica and Greenland sites". In: *Journal of Geophysical Research: Atmospheres* 108.D24.

Herron, M. M. and C. C. Langway (1980). "Firn densification: an empirical model". In: *Journal of Glaciology* 25.93.

Hvidberg, C. S., A. Grinsted, D. Dahl-Jensen, S. A. Khan, A. Kusk, J. K. Andersen, N. Neckel, A. Solgaard, N. B. Karlsson, H. A. Kjar, and P. Vallelonga (2020). "Surface velocity of the Northeast Greenland Ice Stream (NEGIS): Assessment of interior velocities derived from satellite data by GPS". In: *Cryosphere* 14.10, pp. 3487–3502. ISSN: 19940424. DOI: 10.5194/tc-14-3487-2020.

Lipenkov, V. Y. (2018). "How air bubbles form in polar ice". In: *Earth's Cryosphere* 22.2, pp. 16–28. DOI: 10.21782/KZ1560-7496-2018-2.

Mariotti, A. (1983). "Atmospheric nitrogen is a reliable standard for natural 15N abundance measurements". In: *Nature* 303.5919, pp. 685–687.





Mitchell, L. E., C. Buizert, E. J. Brook, D. J. Breton, J. Fegyveresi, D. Baggenstos, A. Orsi, J. Severinghaus, R. B. Alley, M. Albert, R. H. Rhodes, J. R. Mcconnell, M. Sigl, O. Maselli, and S. Gregory (2015). "Observing and modeling the influence of layering on bubble trapping in polar firn". In: *Journal of Geophysical Research : Atmospheres*, pp. 2558–2574. DOI: 10.1002/2014JD022766.Received.

Mojtabavi, S., F. Wilhelms, E. Cook, S. Davies, G. Sinnl, M. Skov Jensen, D. Dahl-Jensen, A. Svensson, B. Vinther, S. Kipfstuhl, G. Jones, N. Karlsson, S. H. Faria, V. Gkinis, H. Kjær, T. Erhardt, S. Berben, K. Nisancioglu, I. Koldtoft, and S. O. Rasmussen (2020). "A first chronology for the East GReenland Ice-core Project (EGRIP) over the Holocene and last glacial termination". In: *Climate of the Past Discussions* 16.December, pp. 2359–2380. ISSN: 1814-9324. DOI: 10.5194/cp-16-2359-2020. URL: https://cp.copernicus.org/articles/16/2359/2020/.

Morcillo, G., S. H. Faria, and S. Kipfstuhl (2020). "Unravelling Antarctica's past through the stratigraphy of a deep ice core: an image-analysis study of the EPICA-DML line-scan images". In: *Quaternary International* February. ISSN: 10406182. DOI: 10.1016/j.quaint.2020.07.011.

Oraschewski, F. M. and A. Grinsted (2021). "Modeling enhanced firn densification due to strain softening". In: *The Cryosphere* September, pp. 1–24.

Petit, J. R., J. Jouzel, D. Raynaud, N. I. Barkov, J.-M. Barnola, I. Basile, M. Bender, J. Chappellaz, M. Davisk, G. Delaygue, M. Delmotte, V. M. Kotlyakov, M. Legrand, V. Y. Lipenkov, C. Lorius, L. Pépin, C. Ritz, E. Saltzmank, and M. Stievenard (1999). "Climate and atmospheric history of the past 420,000 years from the Vostok ice core, Antarctica The recent completion of drilling at Vostok station in East". In: *Nature* 399, pp. 429–436. URL: www.nature.com.

Riverman, K. L., R. B. Alley, S. Anandakrishnan, K. Christianson, N. D. Holschuh, B. Medley, A. Muto, and L. E. Peters (2019). "Enhanced Firn Densification in High-Accumulation Shear Margins of the NE Greenland Ice Stream". In: *Journal of Geophysical Research: Earth Surface* 124.2, pp. 365–382. ISSN: 21699011. DOI: 10.1029/2017JF004604.

Schwander, J., J.-M. Barnola, C. Andrie, M. Leuenberger, A. Ludin, D. Raynaud, and B. Stauffer (1993). "The age of the air in the firn and the ice at Summit, Greenland". In: *Journal of Geophysical Research: Atmospheres* 98.D2, pp. 2831–2838.

Schwander, J. and B. Stauffer (1984). "Age difference between polar ice and the air trapped in its bubbles". In: *Nature* 311.5981, pp. 45–47.

Schwander, J., B. Stauffer, and A. Sigg (1988). "Air mixing in firn and the age of the air at pore close-off". In: *Annals of Glaciology* 10, pp. 141–145.

Sowers, T., M. Bender, D. Raynaud, and Y. S. Korotkevich (1992). "δ15N of N2 in air trapped in polar ice: a tracer of gas transport in the firn and a possible constraint on ice age-gas age differences". In: *Journal of Geophysical Research* 97.D14. ISSN: 01480227. DOI: 10.1029/92jd01297.

Stoll, N., J. Westhoff, P. Bohleber, A. Svensson, D. Dahl-Jensen, C. Barbante, and I. Weikusat (2023). "Chemical and visual characterisation of EGRIP glacial ice and cloudy bands within". In: *The Cryosphere Discussions*, pp. 1–32.

Svensson, A., S. W. Nielsen, S. Kipfstuhl, S. J. Johnsen, J. P. Steffensen, M. Bigler, U. Ruth, and R. Röthlisberger (2005). "Visual stratigraphy of the North Greenland Ice Core Project (NorthGRIP) ice core during the last glacial period". In: *Journal of Geophysical Research: Atmospheres* 110.2, pp. 1–11. ISSN: 01480227. DOI: 10.1029/2004JD005134.

Ueltzhöffer, K. J., V. Bendel, J. Freitag, S. Kipfstuhl, D. Wagenbach, S. H. Faria, and C. S. Garbe (2010). "Distribution of air bubbles in the EDML and EDC (Antarctica) ice cores, using a new method of automatic image analysis". In: *Journal of Glaciology* 56.196, pp. 339–348. ISSN: 00221430. DOI: 10.3189/002214310791968511.

Vallelonga, P. et al. (2014). "Initial results from geophysical surveys and shallow coring of the Northeast Greenland Ice Stream (NEGIS)". In: *Cryosphere* 8.4, pp. 1275–1287. ISSN: 19940424. DOI: 10.5194/tc-8-1275-2014.



Vandecrux, B. et al. (2023). "The historical Greenland Climate Network (GC-Net) curated and augmented Level 1 dataset". In: *Earth System Science Data Discussions* 2023, pp. 1–35. DOI: 10.5194/essd-2023-147. URL: https://essd.copernicus.org/preprints/essd-2023-147/.

Westhoff, J., S. Kipfstuhl, A. Svensson, D. Dahl-Jensen, and I. Weikusat (2019). "Part 1: Visual Stratigraphy of EastGRIP - Holocene". In: *ERDA*. DOI: doi.org/10.17894/ucph.2f43d7c8-ae7f-47af-ad8a-4aaab2784b87.

Westhoff, J., G. Sinnl, A. Svensson, J. Freitag, H. A. Kjær, P. Vallelonga, B. Vinther, S. Kipfstuhl, D. Dahl-Jensen, and I. Weikusat (2022). "Melt in the Greenland EastGRIP ice core reveals Holocene warm events". In: *Climate of the Past* 18.5, pp. 1011–1034. DOI: 10.5194/cp-18-1011-2022. URL: https://cp.copernicus.org/articles/18/1011/2022/.

Westhoff, J., N. Stoll, S. Franke, I. Weikusat, P. Bons, J. Kerch, D. Jansen, S. Kipfstuhl, and D. Dahl-Jensen (2020). "A Stratigraphy Based Method for Reconstructing Ice Core Orientation". In: *Annals of Glaciology* 62.85-86, pp. 191–202. DOI: 10.1017/aog.2020.76.

Witrant, E., P. Martinerie, C. Hogan, J. C. Laube, K. Kawamura, E. Capron, S. A. Montzka, E. J. Dlugokencky, D. Etheridge, T. Blunier, and W. T. Sturges (2012). "A new multi-gas constrained model of trace gas non-homogeneous transport in firn: Evaluation and behaviour at eleven polar sites". In: *Atmospheric Chemistry and Physics* 12.23, pp. 11465–11483. ISSN: 16807316. DOI: 10.5194/acp-12-11465-2012.