# Peer review of "Combining traditional and novel techniques to increase our understanding of the lock-in depth of atmospheric gases in polar ice cores - results from the EastGRIP region"

_EGUsphere, 2023_

## Referee Comment (RC1)

Review for *Combining traditional and novel techniques to increase our understanding of the lock-in depth of atmospheric gases in polar ice cores - results from the EastGRIP region* by Westhoff et al.

General Comments:

The authors present and interpret new optical data to investigate the LID and bubble close-off in an EastGRIP ice core. The method is novel, and the data could be the basis for a strong paper. Nonetheless, I have significant concerns about both the organization and the strength/clarity of the scientific reasoning presented in the manuscript.

One major concern is that the manuscript focuses primarily on the OLID, but the scientific motivation for determining a specific OLID is not clear. The results plotted in Figure 2d and f and Figure 7 have more importance in the broader scientific context of understanding bubble close-off and delta age, given the obvious shortcomings of the Goujon/Barnola parameterization. If the OLID is not the actual motivation, the authors might consider broadening the focus to the depth-range of bubble close-off and the associated implications for delta age.

A second major concern is that the exact methodology for finding the OLID and the "bubble proxy" is poorly explained. As the authors state, this is a new methodology, and it needs to be very clear to the reader. Sections 3.2-3.3 are spent describing and interpreting data from a new method, which the reader has no way of understanding. The information in section 3.4 and Appendices A-D should be given before the data is presented and interpreted.

Specific Comments:

Section 1: The scientific motivation and larger context for this study are poorly defined. Namely, why pursue an optical method for determining the LID? Or, stated another way, what do we learn from 58.3 m that we didn't learn from 58-61 m? Does it provide more precise information about the delta age? If so, more time should be spent discussing the importance of delta age and the physical site characteristics that control it. Section 1.3 is labeled "Motivation," but it summarizes the paper rather than providing scientific motivation.

Section 2.1: This section should be combined with 1.2 and should probably come after the Introduction.

It's not clear which sites are affected by compaction due to flow.

An additional note: Here and throughout the paper, EastGRIP and S6 seem to be used interchangeably. (For example, Figure 2e is labeled EastGRIP open porosity, but I assume it is the same firn air pumping campaign from S6 that is plotted in 2a and 2b). Please clarify throughout the paper

Figure 2: please comment on the data gap between 72 and 75 m in 2c and 2d.

Section 2.3.4: Did the authors consider trying the Mitchell et al., 2015 parameterization, which has a more gradual bubble close-off? How realistic is the modified porosity profile relative to other measurements and parameterizations? The tracers used to tune the model should also be clarified.

Section 3: Overall, the clarity of the scientific reasoning in this section needs improvement.

Section 3.1: It is not clear whether these measurements were made as a part of this research or if they are previously published. If they are previously published measurements, they should be cited.

Section 3.2.1-3.3: The information in section 3.4 and Appendices B-C is necessary to understand these sections. Please reorganize.

The geometry of the 1x5 cm$^2$ relative to the 165 cm slab is not clear. A figure similar to 2h in Westhoff et al. (2020) would be clarifying.

The "bubble proxy" is not explained:

1) Does one bright spot correspond to one bubble?
2) How is a "bright spot?" defined? Why use one pixel cutoff value versus another?
3) Can a single bright spot be more than one pixel?
4) Can a bubble be more than one pixel?
5) Is the proxy qualitative or quantitative?
6) Is the basis of the proxy empirical or theoretical? If theoretical, section 3.4 needs additional details.

Figure 5b: please explain the different pixel cutoff values. It appears that the results are quite sensitive to the choice of 60.

3.3 is labeled "Density and Visual Stratigraphy derived lock-in depth," and the authors seem to infer that the density measurements suggest the LID is the 58.3 m layer but L162 states: "Between 55 and approximately 67 m depth, density values lie between 790kg/m3 and 830kg/m3, suggesting this to be the LIZ." Please clarify.

Section 3.4: This section is the crux of the methodology/proxy and therefore needs more scientific justification. It also needs citations. It is not enough to draw a picture of what may be happening without explaining the underlying optical physics:

1) Why is it only closed spherical bubbles that make bright spots? What about mostly spherical pores that aren't completely closed off? It seems like any curved air/ice interface could potentially act as focusing lens if it is oriented correctly?
2) Section 3.2.1 describes light "reflection, refraction, and scattering" but Figure 6 shows light refracting and focusing. Please clarify.
3) What is happening with the closed pores between 50 and 58.3 m? Are the closed pores in that depth interval "odd shaped" and they ultimately evolve towards spherical?
4) What is the evidence for closed pores at 50 m? It does not appear to be from the line scan data.

It seems like a melt layer such as the one mentioned in 3.1 would make an effective impermeable layer, potentially preventing diffusion without producing any bright spots. Please address this.

Section 3.5:

The relationship between layering, bubble close-off, and the LID Is not clearly explained. Please clarify.

The text implies that the presence of a single bright spot is evidence that a layer is impermeable. But, if a single bright spot corresponds to a single bubble, that does not make sense. Please clarify.

"Some layers have bright spots, while others appear darker (d), going hand in hand with the number of bubbles we find (e)."- e) only shows number of bright spots… please explain what is meant by "number of bubbles you find" Is it just the number of bright spots? Do the maximums in pixel value correlate to maximums in bright spots? If so, please make a plot that shows the covariance or do a statistical test because it is not obvious from Figure 5d-e. Please clarify.

Section 3.6.1- The information in appendix D is necessary to understand this section. Please reorganize.

It needs to be mentioned these calculations are done using closed porosity from parameterizations, not data here.

Section 3.6.2

This section is potentially useful for improving understanding of delta age and the age distribution of air trapped in polar ice. I recommend a more detailed discussion here.

Can you show the Mitchell parameterization of Figure 7? Or even better implement it in the firn air model?

Section 3.7

The authors state that there is a correlation between closed pore space and image brightness, but it is not clear where the information about closed pore-space is coming from unless it is the image brightness. Please clarify.

Section 4

This should just be "10-50" years, not "± 10-50 years." Moreover, the mixing delay should be easy to calculate with the firn air model. Why not use that instead of Schwander's "typical number?"

Additionally, the authors state earlier in the paper that some closed porosity forms as shallow as 50 m. This is not accounted for in paragraph 2. Please address.

Section 5- please rephrase this first sentence "It is important to…"

Other stylistic notes:

Phrases like "hand in hand" (L243) and "has been around for a long time" (L230) are not appropriate for a scientific manuscript. I suggest something like "Layering in the LIZ can influence bubble closure (Blunier et al., 2000; Fourteau et al., 2019)" and "The maximum pixel brightness covaries with the number of bright spots" at those lines.

In general, please carefully proofread the grammar and give some careful thought to phrasing. There are many opportunities to make the writing clearer and more concise. For example, L296-297 could be revised to:

"All the data presented in this work indicate that the transition between the diffusive and non-diffusive zone in the EastGRIP area occurs between 58 m and 61 m."

And line 283-284

"Density measurements and visual stratigraphy data can reveal more details about the firn-ice transition."

Please clean up figure axes and axes labels. Some labels are left justified, and some are centered. Centered is best throughout.

---

## Author Comment (AC1)

**Review 1**

Review for *Combining traditional and novel techniques to increase our understanding of the lock-in depth of atmospheric gases in polar ice cores - results from the EastGRIP region* by Westhoff et al.

**General Comments:**

The authors present and interpret new optical data to investigate the LID and bubble close-off in an EastGRIP ice core. The method is novel, and the data could be the basis for a strong paper. Nonetheless, I have significant concerns about both the organization and the strength/clarity of the scientific reasoning presented in the manuscript.

One major concern is that the manuscript focuses primarily on the OLID, but the scientific motivation for determining a specific OLID is not clear. The results plotted in Figure 2d and f and Figure 7 have more importance in the broader scientific context of understanding bubble close-off and delta age, given the obvious shortcomings of the Goujon/Barnola parameterization. If the OLID is not the actual motivation, the authors might consider broadening the focus to the depth-range of bubble close-off and the associated implications for delta age.

A second major concern is that the exact methodology for finding the OLID and the "bubble proxy" is poorly explained. As the authors state, this is a new methodology, and it needs to be very clear to the reader. Sections 3.2-3.3 are spent describing and interpreting data from a new method, which the reader has no way of understanding. The information in section 3.4 and Appendices A-D should be given before the data is presented and interpreted.

*Thank you for the review of our manuscript. You have pointed out many valuable points that will help improve the manuscript. We will implement your review into the new version of the manuscript.*

*The major objective of the manuscript is to describe and evaluate a new experimental method to locate firn bubble close-off. New direct experimental constraints are especially useful for a complex site that challenges model limitations. The EGRIP site is affected by strong horizontal ice flow influencing firn densification and structure, as well as climate change further inducing time variations in firn temperature and snow accumulation. Such processes are not represented in a 1D steady-state model. In this non-steady state firn context, delta-age is likely to have changed in the recent past, however, estimates of delta-age and age distributions in ice will be provided. An alternative to the Goujon/Barnola parameterization (Mitchell et al., 2015) has been tested as suggested, and results will be provided in Figures 2 and 7.*

*We will restructure section 3, as mentioned here and in the specific comments. We thereby hope to make the structure of the manuscript better.*

**Specific Comments:**

Section 1: The scientific motivation and larger context for this study are poorly defined. Namely, why pursue an optical method for determining the LID? Or, stated another way, what do we learn from 58.3 m that we didn't learn from 58-61 m? Does it provide more precise information about the delta age? If so, more time should be spent discussing the importance of delta age and the physical site characteristics that control it. Section 1.3 is labeled "Motivation," but it summarizes the paper rather than providing scientific motivation.

*We will rework the introduction and the motivation. We will lay a larger focus on the delta age and what we can gain from the different methods.*

Section 2.1: This section should be combined with 1.2 and should probably come after the Introduction.

*We will rearrange the beginning of the paper. Motivation will be moved forward as the new section 1.2. Then Site locations and variations between cores will be combined thereafter.*

It's not clear which sites are affected by compaction due to flow.

*We will clarify that all cores are affected by flow, except NEEM.*

An additional note: Here and throughout the paper, EastGRIP and S6 seem to be used interchangeably. (For example, Figure 2e is labeled EastGRIP open porosity, but I assume it is the same firn air pumping campaign from S6 that is plotted in 2a and 2b). Please clarify throughout the paper

*That is a very valid point and will be addressed throughout the paper. In the discussion section, where we assume similar features for both cores, we will clarify this.*

Figure 2: please comment on the data gap between 72 and 75 m in 2c and 2d.

*We will include that the ice quality was too poor on that section for the gas CFA methodology, with 7 to 8 breaks along a CFA stick.*

Section 2.3.4: Did the authors consider trying the Mitchell et al., 2015 parameterization, which has a more gradual bubble close-off? How realistic is the modified porosity profile relative to other measurements and parameterizations? The tracers used to tune the model should also be clarified.

*Thank you for suggesting the use of Mitchell et al., 2015 parameterization. We implemented it in its "bulk" form (Equations 6 to 9) and it led to an improved match with the upper part of the methane data in the closed porosity. The modified porosity tested in the manuscript was not meant to be optimal but to illustrate the impact of the chosen parameterization. Figures 2 and 7 will be modified to include these results. The tracers used to tune the diffusivity in the open porosity are: $CH_4$, $SF_6$, CFC-12, CFC-113, $CH_3CCl_3$, and HFC-134a. Using only methane leads to nearly the same results. It will be clarified in the manuscript.*

Section 3: Overall, the clarity of the scientific reasoning in this section needs improvement.

*We will restructure the section, add a figure for clarification, and increase scientific reasoning.*

Section 3.1: It is not clear whether these measurements were made as a part of this research or if they are previously published. If they are previously published measurements, they should be cited.

*They were not made in direct connection to this research, but have also not been published before. We will include the information.*

Section 3.2.1-3.3: The information in section 3.4 and Appendices B-C is necessary to understand these sections. Please reorganize.

*We agree that these sections need reorganization. Section 3.4. will be moved forward to better introduce the visual effect of bubbles on the line scan images. A short version of appendix B and C will be added to the main text and the reference to more details in the appendix will be made clear.*

The geometry of the 1x5 cm2 relative to the 165 cm slab is not clear. A figure similar to 2h in Westhoff et al. (2020) would be clarifying.

The "bubble proxy" is not explained:

1) Does one bright spot correspond to one bubble?

2) How is a "bright spot?" defined? Why use one pixel cutoff value versus another?

3) Can a single bright spot be more than one pixel?

4) Can a bubble be more than one pixel?

5) Is the proxy qualitative or quantitative?

6) Is the basis of the proxy empirical or theoretical? If theoretical, section 3.4 needs additional details.

*Thank you for pointing this out. We will create a figure accommodating the comments above. We will also make sure to elaborate on this clearly in the text.*

Figure 5b: please explain the different pixel cutoff values. It appears that the results are quite sensitive to the choice of 60.

*The choice of 60 is arbitrary and based on what appears bright by the eye. Similar results were obtained for values of 50 or 70, but not shown here. The values of 150 and 250 show similar trends, yet with a much lower amplitude, not making them feasible for a visually appealing plot. This will be explained in the manuscript and also implemented into the figure from the comment above.*

3.3 is labeled "Density and Visual Stratigraphy derived lock-in depth," and the authors seem to infer that the density measurements suggest the LID is the 58.3 m layer but L162 states: "Between 55 and approximately 67 m depth, density values lie between 790kg/m3 and 830kg/m3, suggesting this to be the LIZ." Please clarify.

*This is indeed misleading. The density part will be removed from the title. The LIZ ranges over the depth from 55 to 67 m. The specific depth derived from the visual stratigraphy is 58.3 m. Which is also confirmed by the density measurements. This will be clarified in the next version of the manuscript.*

Section 3.4: This section is the crux of the methodology/proxy and therefore needs more scientific justification. It also needs citations. It is not enough to draw a picture of what may be happening without explaining the underlying optical physics:

1) Why is it only closed spherical bubbles that make bright spots? What about mostly spherical pores that aren't completely closed off? It seems like any curved air/ice interface could potentially act as focusing lens if it is oriented correctly?

2) Section 3.2.1 describes light "reflection, refraction, and scattering" but Figure 6 shows light refracting and focusing. Please clarify.

3) What is happening with the closed pores between 50 and 58.3 m? Are the closed pores in that depth interval "odd shaped" and they ultimately evolve towards spherical?

4) What is the evidence for closed pores at 50 m? It does not appear to be from the line scan data.

*Citations will be added to better the underlying optical physics. We will refer to the shapes and their effect on the line scan images separately and also clarify the wording of reflection, focussing, etc. The questions from 1) and 3) will be addressed in the manuscript. 4) is answered in the appendix and will be included into the main section of the manuscript. The entire section 3.4 will be moved forward to better explain the visual stratigraphy in sections 3.2 and 3.3.*

It seems like a melt layer such as the one mentioned in 3.1 would make an effective impermeable layer, potentially preventing diffusion without producing any bright spots. Please address this.

*This is very true and will be included in the discussion. The melt layer could create an impermeable layer and thus induce the lock-in, yet this one does not coincide with the appearance of bubbles and results from the firn air pumping. We will elaborate on this in the discussion.*

Section 3.5:

The relationship between layering, bubble close-off, and the LID Is not clearly explained. Please clarify.

The text implies that the presence of a single bright spot is evidence that a layer is impermeable. But, if a single bright spot corresponds to a single bubble, that does not make sense. Please clarify. "Some layers have bright spots, while others appear darker (d), going hand in hand with the number of bubbles we find (e)."- e) only shows number of bright spots… please explain what is meant by "number of bubbles you find" Is it just the number of bright spots? Do the maximums in pixel value correlate to maximums in bright spots? If so, please make a plot that shows the covariance or do a statistical test because it is not obvious from Figure 5d-e. Please clarify.

*Thank you for pointing this out. It is indeed not clear and we will include a figure for clarification and address the points you mentioned.*

Section 3.6.1- The information in appendix D is necessary to understand this section. Please reorganize.

It needs to be mentioned these calculations are done using closed porosity from parameterizations, not data here.

*We will mention that this is calculated from closed porosity parametrization.*
*For a better understanding of the section, we will include appendix D into the text. As also mentioned in the comments further above, restructuring of section 3 is necessary and will be done.*

Section 3.6.2

This section is potentially useful for improving understanding of delta age and the age distribution of air trapped in polar ice. I recommend a more detailed discussion here.

Can you show the Mitchell parameterization of Figure 7? Or even better implement it in the firn air model?

*The discussion will be extended to increase the focus on delta age and age distribution. It will also be mentioned more clearly in the motivation. Delta age distributions and mean values obtained with all tested parameterizations, including Mitchell et al. 2015, will be provided.*

Section 3.7

The authors state that there is a correlation between closed pore space and image brightness, but it is not clear where the information about closed pore-space is coming from unless it is the image brightness. Please clarify.

*We will elaborate further on this, as it is solely from the image brightness.*

Section 4

This should just be "10-50" years, not "± 10-50 years." Moreover, the mixing delay should be easy to calculate with the firn air model. Why not use that instead of Schwander's "typical number?" Additionally, the authors state earlier in the paper that some closed porosity forms as shallow as 50 m. This is not accounted for in paragraph 2. Please address.

*We will remove the plus-minus, clarify the "as shallow as 50m", and add the following:*
*From the porosity parametrization, we have three delta-age probability distributions: blue 344 years, green 315 years, and black 337 years (fig. X). The results are dependent on the model limitations, e.g. steady-state 1D  model with fixed accumulation rate and temperature, no layering, no flow-related thinning, etc.).*

Section 5- please rephrase this first sentence "It is important to…"
Other stylistic notes:
Phrases like "hand in hand" (L243) and "has been around for a long time" (L230) are not appropriate for a scientific manuscript. I suggest something like "Layering in the LIZ can influence bubble closure (Blunier et al., 2000; Fourteau et al., 2019)" and "The maximum pixel brightness covaries with the number of bright spots" at those lines.
In general, please carefully proofread the grammar and give some careful thought to phrasing. There are many opportunities to make the writing clearer and more concise. For example, L296-297 could be revised to:
"All the data presented in this work indicate that the transition between the diffusive and non-diffusive zone in the EastGRIP area occurs between 58 m and 61 m."
And line 283-284
"Density measurements and visual stratigraphy data can reveal more details about the firn-ice transition."
*Thank you for the suggestions, they will be implemented. We will also review the manuscript for clearer and more concise language.*

Please clean up figure axes and axes labels. Some labels are left justified, and some are centered. Centered is best throughout.
*Axes and labels will be cleaned up and centered in a consistent manner.*

---

## Author Comment (AC2)

**Review 2**

**General review:**

The article addresses the intricacies surrounding bubbles and LIZ formation within the firn layer of the Greenland ice sheet, employing both traditional and innovative optical methodologies. This research holds significant importance in advancing our understanding of gas ages and their associated smoothing effects. However, the previous conventional methods fell short in capturing the the subtle intricacies of LIZ formation. The authors have introduced a novel term, the optical lock-in depth (OLID), and conducted a comparative analysis with conventional approaches. These novel findings significantly augment our understanding, and the interpretations offered are judicious. Nonetheless, the methods elucidated poses challenges in comprehension, necessitating structural and terminological refinements to enhance clarity. Editorial revisions are imperative prior to publication.

*Thank you for the review of our paper. We do not have any objections to any of the review comments and will implement them into the new version of the manuscript.*

**Specific comments:**

Line 20: What does $\delta^{15}N$ signify?  Is it $\delta^{15}N$ of $N_2$? Please specify.

*We will add information to lines 19-20 to increase clarification: Blunier et al. (2000) describe three zones derived from the gravitational settling of δ15N of N2: an upper convection zone, a diffusive zone, and a lock-in zone (LIZ).*

Page 2, Move '1.3 Motivation' before '1.2 Site Locations' for better flow.

*Will be done and the text adjusted. We will also combine the sections "variations between cores" and "site locations" for better readability.*

Page 2, In section 1.1 or 1.3, describe the issues with previous conventional methods.

*We will enhance the emphasis on the new method and its relevance:*
*New direct experimental constraints are especially useful for a complex site that challenges model limitations. The EGRIP site is affected by strong horizontal ice flow influencing firn densification and structure, as well as climate change further inducing time variations in firn temperature and snow accumulation.*

Line 32: Can you provide coordinates for the S2 location?

*Will be provided for the revised manuscript.*

Figure 1c caption: why is the photo from Little Dome C shown? Did you use the same equipment? Please clarify this choice.
*Yes, the same equipment was used. We will clarify this in the text.*

Lines 48 and 50: should not be broken into separate paragraphs.
*This will be changed.*

Line 59: Specify what you mean by "issues".
*Will be added: … issues, such as leakages in the system or sucking air from other stratigraphic layers, …*

Line 81: absence of gravitational enrichment => absence of further gravitational enrichment?
*Correct – will be changed accordingly.*

Figure 2b caption: transition depth of '58 m'? It is not likely to be about 60 m. Refer to text at Line 96.
*58 m is the change of slope, between the two values.*

Figure 2f caption: Witrant et al. 2012) => Witrant et al. (2012)
*This will be corrected.*

Line 96: Explain why the top of the LID is differently defined using data of $\delta^{15}$N-N$_2$ and CO$_2$ concentration.
*This difference is illustrated in a multi-sites perspective in Table 2 of Witrant et al. (2012). In terms of gas transport, on one hand CO$_2$ is more affected by diffusion than gravitation due to its important atmospheric time trend which induces a strong concentration gradient between the top and the bottom of the firn that diffusion tends to reduce. On the other hand $\delta^{15}$N-N$_2$ is more weakly affected by diffusion due to the absence of atmospheric time trend and relatively more affected by gravitation as well as advection due to the sinking of firn layers. Together with the site-dependent abruptness of gas transport reduction around the LID, differences in the behaviors of CO$_2$ and $\delta^{15}$N-N$_2$ may occur. They may also respond differently to time variations in firn temperature or snow accumulation.*

Line 104~107: Add more reference. Similar features were reported also in Mitchell et al. (2015) and Jang et al. (2019)
*References will be added.*

Line 108: Erase '(fig. 2c)'
*We will erase the double mentioning of figure 2c.*

Line 127: Even the green line in fig. 2d does not well matched with the CH4 concentrations in high density layers (minima of $CH_4$ concentration). The authors may suggest plausible reasons.

*The model limitations for simulating the complex EGRIP site and influence of the open/closed porosity parameterization will be further discussed (see also answers to first review).*

Line 156-158: Mention other definitions of the close-off zone. Consider citing Martinerie et al. (1992).

*Thank you for suggesting this, we will add the citation.*

Line 161: Between 55 and... => Between 50(?) and...

*That is correct and we will correct the value.*

Line 165-174: Specify that details of observations are described in Appendix C.

*Will be explained in the main text and we will point towards appendix C.*

Figure 4 caption & Line 181: please define 'pixel value'

*Thank you for noticing. This is clearly missing and will be explained.*

Figure 5a caption: please address Appendix B if it is related to the method.

*The relation will be established.*

Line 221: Define 'percolation transition'

*A reference will be added and a short explanation: the percolation transition is the transition of small and disconnected clusters merging into larger and connected clusters (Li et al., 2021).*

Line 227: Specify 'clusters'

*Clusters of bubbles (will be added)*

Line 231: add Jang et al. (2019) to the references

*Will be done*

Line 241: Define 'maximum pixel value'

*Reference to "maximum" in fig. 2a will be added for clarification and a short comment included. .*

Line 273-274: Address 'Appendix D' where the 'pixel values above 20' are described.

*We will address the appendix and include more of it in the main text.*

Line 281: 55 to 58 m depth => 55 to 58 m depth?
*Correct, "55 to 60" will be changed to "55 to 58".*

Line 298-299: Change 'agree very well' to 'agree well'. We see a difference in the LIDs defined by $CO_2$ and $\delta^{15}N\text{-}N_2$
*Will be changed.*

Line 325-336: Relocate and shorten 'Appendix A' in the main text's site description section.
*We will include Appendix A into the main text.*

Figure D1b: Specify the meaning of 'gray value' on the y-axis label? Is it the pixel value?
*Will be changed to pixel value for consistency.*

Line 432: Erase 'Appenix:'
*Will be erased*

**Refercence:**
Martinerie, P., Raynaud, D., Etheridge, D. M., Barnola, J. M., and Mazaudier, D.: Physical and Climatic Parameters which Influence the Air Content in Polar Ice, **Earth Planet. Sc. Lett**., 112, 1–13, https://doi.org/10.1016/0012-821X(92)90002-D, (1992)
Mitchell, L., Christo Buizert, Edward Brook, Daniel Breton, John Fegyveresi, Daniel Baggenstos, Anais Orsi, Jeffrey Severinghaus, Richard B. Alley, Mary Albert, Rachael H. Rhodes, Joseph R. McConnell, Michael Sigl, Olivia Maselli, Stephanie Gregory and Jinho Ahn. Observing and modelling the influence of layering on bubble trapping in polar firn, **Journal of Geophysical Research**, 120, doi:10.1002/2014JD022766 (2015)
Youngjoon Jang, Sang Bum Hong, Christo Buizert, Hun-Gyu Lee, Sang-young Han, Ji-Woong Yang, Yoshinori Iizuka, Akira Hori, Yeongcheol Han, Seong Joon Jun, Pieter Tans, Taejin Choi, Seong-Joong Kim, Soon Do Hur and Jinho Ahn, Very old firn air linked to strong density layering at Styx Glacier, coastal Victoria Land, East Antarctica, **The Cryosphere, 13, 2407-2419** (2019)

---

## Author Comment (AC3)

**Editor / Review 3**

In addition to the posted two reviews I have a comment from another reviewer, Jeff Severinghaus, which I will paste below as there was a complication getting this in the system. This review is very positive and just makes some minor comments for the authors to consider.

Review of "Combining traditional and novel techniques to increase our understanding of the lock-in depth of atmospheric gases in polar ice cores – results from the EastGRIP region"

Authors: J. Westhoff et al.                                                                    March 20, 2024

This manuscript presents a novel and very interesting new method of determining the depth in an ice core at which air bubbles are effectively closed off, taking advantage of the optical properties of bubbles and ice.  In polar settings, slow densification of the firn (porous ice) typically leads to formation of ice with trapped air bubbles after several hundred years.

In detail, the authors show that "bright spots" appear in the optical images around 61 m depth, very close to the depth that is found via classical methods for finding the depth of the bubble close-off.   The authors make a convincing case that these "bright spots" have a mechanistic linkage to the ice properties at around 61 m depth.  With their new optical methods, the authors show persuasively that the "bright spots" record a critical change in the firn geometry and structure.

They proposed that the "bright spots" are due to the creation of bubbles with semi-spherical geometry, which is known to produce bright reflections  As such, the "bright spots" are shown to have a mechanistic and meaningful origin that can be further probed in the future. The authors also use traditional methods such as density measurements to verify the closure of bubbles at this depth.  Overall, this is an excellent contribution to our understanding of the firn-to-ice transition in polar ice sheets, and represents a new and valuable metric of where the "classical bubble close-off depth" happens.  I recommend that this work be published with only minor edits, as spelled out below.
 Jeff Severinghaus, reviewer

*Thank you for your very positive review. We will include your comments and suggestions for changes throughout the manuscript.*

Minor and editorial comments:

Line 56  this would be made clearer for the reader by writing "inserted an inflatable rubber bladder"
*We will add the suggestion for more clarity.*

Line 57  perhaps this should be ¼ inch, not 1.4 inch?  Normally the purge line is about 3/8" to ½" in diameter, and the air sampling line is  ¼"
*Thanks for pointing this out, yes it is ¼ inch. We will correct this.*

Line 59  say "were monitored on-site during pumping to detect contamination issues…"
*This will be changed.*

Line 64 "and diffusive mixing in the vertical direction essentially stops"   [This is an important distinction because many studies have shown that horizontal diffusion and advection can remain prevalent due to horizontal high-permeability layers (typically summer layers) even when vertical air flow is completely shut off]
*Thanks for the note.*

Line 72  "evolution of pore closure"
*Will be corrected*.

Line 78  for the reader, it might help to cite a ref here at 2), since thermal fractionation isn't widely known in the community.  You could cite Severinghaus et al., 1998 Nature
*We will include the citation to make it more clear.*

Line 79  your interpretation of an extremum in 15N at 12 m from the previous winter's cold is mistaken  -  in fact the extremum at 12 m is due to the recent summer atmospheric warmth just months before the pumping.  The reason is that the atmosphere cannot change its 15N, due to its virtually infinite reservoir, so the nitrogen gas in the top dozen meters of firn (which is colder than the atmosphere in local summer) must become fractionated with the heavy isotope 15N becoming enriched.  The signal of the previous winter, on the other hand, shows at 18 m, with a slightly depleted 15N.  See Severinghaus et al. 2001 (G Cubed) for a more complete explanation of the phenomenon, including a wintertime firn air sampling at South Pole along with the usual summertime sampling of firn air.  As expected, the top 12 meters

of firn shows very negative 15N in winter (because the atmosphere is so much colder than the air in the firn).
*Thermal fractionation of air in polar firn by seasonal temperature gradients – G-Cubed   J. P. Severinghaus, A. Grachev, M. Battle   2001*
*Thank you for the correction and for pointing this out. We will add the reference and adjust the text accordingly.*

Line 84  The slight decrease of 15N with increasing depth within the lock-in zone is well known to be due to global warming and resulting firn thermal fractionation over the past 4 to 5 decades, not to contamination.  This effect has been extensively documented by Orsi et al., 2017.  You should cite her work: *The recent warming trend in North Greenland ,  Geophysical Research Letters, AJ Orsi et al., 2017*
*Thank you also for this input. We will adjust the text accordingly and include the correction and reference.*

Line 91  Check your calculation of "almost 400 years of snow accumulation" in 66 m.  It's probably more like 330 years.   You have to take account of the fact that annual layers are quite thick in the upper part of the firn, with snow densities of only 0.35 to 0.55 kg per liter, in comparison to snow densities of 0.83 to 0.84 kg per liter in the lock-in zone.
*The age at 66 m depth is 385 years b2k (Mojtabavi et al. 2020). We will add the reference to the age.*

Line 109  Fix the statement "around 66 m the bubbles are essentially closed off".  This is not consistent nor accurate, since the same sentence states that "layers with not fully closed pores can be found down to 71.5 m".
*Will be changed to: "Layers with not fully closed pores can be found down to 71.5 m although many pores are already closed at 66 m depth."*

Line 135  "11 cm annual layer thickness"  This doesn't seem right – please check
*According to Mojtabavi et al. (2020) one bag (55cm) in depth contains 5 years, which is roughly 11 cm layer thickness.*

Line 213  It would be helpful to include a reference here on "percolation transitions" – which some readers might not be familiar with.
*A reference will be added and a short explanation: the percolation transition is the transition of small and disconnected clusters merging into larger and connected clusters (Li et al., 2021).*

Line 235  please provide a reference to "strangulation" – it is the first time this word is used in the paper, and many readers will not know what this means.

*Thank you for noticing, we will clarify this, e.g.: "...strangulation, reduction of gas exchange between layers, ..."*

---

## Author Response (AR1)

Dear all,

Thank you for the review of our manuscript.

We have implemented all the suggestions from the reviewers. You can track these in a thorough point-by-point report, submitted as comments to the reviewers.

We have also attached a pdf which indicates all the changes made.

We hope this satisfies the reviewers and editor and are open to further feedback.

All the best,

Julien Westhoff et al.